# Seroepidemiological investigation of Crimean Congo hemorrhagic fever virus in livestock in Uganda, 2017

Luke Nyakarahuka[1,2]*, Jackson Kyondo[1], Carson Telford[3], Amy Whitesell[3], Alex Tumusiime[1], Sophia Mulei[1], Jimmy Baluku[1], Caitlin M. Cossaboom[3], Deborah L. Cannon[3], Joel M. Montgomery[3], Julius J. Lutwama[1], Stuart T. Nichol[3], Stephen K. Balinandi[1], John D. Klena[3], Trevor R. Shoemaker[3]

1 Department of Arbovirology, Emerging and Reemerging Infectious Diseases, Uganda Virus Research Institute, Entebbe, Uganda, 2 Department of Biosecurity, Ecosystems and Veterinary Public Health, College of Veterinary Medicine, Animal Resources and Biosecurity, Makerere University, Kampala, Uganda, 3 Viral Special Pathogens Branch, Division of High-Consequence Pathogens and Pathology, United States Centers for Disease Control and Prevention, Atlanta, Georgia, United States of America

* nyakarahuka@gmail.com

**Data Availability Statement:** Data is available with restriction because the some components of dataset will be used to develop risk maps for rift valley fever in Uganda. These include GIS

## Abstract

Crimean-Congo Hemorrhagic fever (CCHF) is an important zoonotic disease transmitted to humans both by tick vectors and contact with fluids from an infected animal or human. Although animals are not symptomatic when infected, they are the main source of human infection. Uganda has reported sporadic human outbreaks of CCHF in various parts of the country since 2013. We designed a nationwide epidemiological study to investigate the burden of CCHF in livestock. A total of 3181 animals were sampled; 1732 cattle (54.4%), 1091 goats (34.3%), and 358 sheep (11.3%) resulting in overall livestock seropositivity of IgG antibodies against CCHF virus (CCHFV) of 31.4% (999/3181). Seropositivity in cattle was 16.9% and in sheep and goats was 48.8%. Adult and juvenile animals had higher seropositivity compared to recently born animals, and seropositivity was higher in female animals (33.5%) compared to male animals (24.1%). Local breeds had higher (36.8%) compared to exotic (2.8%) and cross breeds (19.3%). Animals that had a history of abortion or stillbirth had higher seropositivity compared to those without a history of abortion or stillbirth. CCHFV seropositivity appeared to be generally higher in northern districts of the country, though spatial trends among sampled districts were not examined. A multivariate regression analysis using a generalized linear mixed model showed that animal species, age, sex, region, and elevation were all significantly associated with CCHFV seropositivity after adjusting for the effects of other model predictors. This study shows that CCHFV is actively circulating in Uganda, posing a serious risk for human infection. The results from this study can be used to help target surveillance efforts for early case detection in animals and limit subsequent spillover into humans.

component and and herd level prevalence as we shall be referring to this publication. However, the data can be availed upon request Uganda Virus Research Institute (UVRI) Ethics and Research Committee that approved this study at email uvrirec@uvri.go.ug

**Funding:** This study was funded by United States Centers for Disease Control and Prevention (CDC) through the National Center for Emerging and Zoonotic Infectious Disease (NCEZD Grant number RFA-CK-13-001. The funders had no role in study design, data collection and analysis, decision to publish, or preparation of the manuscript.

**Competing interests:** The authors have declared that no competing interests exist

## Introduction

Crimean Congo hemorrhagic fever (CCHF) is caused by a single-stranded RNA Crimean-Congo Hemorrhagic Fever orthonairovirus (CCHFV) in the family *Nairoviridae*, order *Bunyavirales* [1]. CCHFV is a zoonotic infection of animals and humans and is of great public health importance [2]. The virus is mainly transmitted by ticks to wild animals and livestock. Most human infections are acquired through contact with infected body fluids of livestock [3], however, humans can also be directly infected through a tick bite [4]. Infected animals do not show overt clinical symptoms, but infected livestock can have a mild fever and viremia enough to cause transmission to vectors and humans [5]. The disease can be severe in humans and cause hemorrhagic manifestations, hence its classification as a viral hemorrhagic fever [6]. CCHF cannot easily be differentiated clinically from other more common tropical infectious diseases including malaria, typhoid, brucellosis and others [7]. Because of this, there is a risk of misdiagnosis without laboratory testing; especially for local health facilities, as CCHF patients can be coinfected with malaria [8, 9]. Uganda had never reported cases of CCHF in humans until the Uganda Virus Research Institute's (UVRI) Viral Hemorrhagic Fever Program in Entebbe initiated surveillance activities for CCHF and other viral causes of hemorrhagic fever [10]. CCHF was first reported in the Northern Ugandan District of Agago in 2013, by three individuals who had slaughtered cattle. Subsequent investigations into this outbreak revealed the presence of CCHFV-specific IgG antibodies in animals and found that the risk of exposure in humans was associated with tick bites and exposure to infected animal body fluids [11]. Concurrently, another outbreak was confirmed in the southern districts of Wakiso and Kiboga, where investigations identified active CCHFV in *Rhipicephalus* spp. ticks via RT-qPCR and CCHFV-specific antibodies in 12% of domesticated ungulates sampled [12]. By 2021, 33 human outbreaks had been reported throughout Uganda, especially in the Cattle Corridor districts, with an overall case fatality rate of 33% [12–14]. High-risk groups for infection include abattoir workers, livestock handlers, butchers, or other occupations requiring handling of domestic livestock [12]. Investigations around these outbreaks of human infection have revealed high IgG seropositivity in livestock, especially in animals from farms linked to confirmed human cases. However, no comprehensive countrywide study has been performed to investigate the burden of CCHFV in livestock. The main objective of this study was to determine the IgG seropositivity towards CCHFV in livestock (cattle, sheep, and goats) from different ecological zones in Uganda and assess the risk factors for CCHFV seropositivity in livestock.

## Methods

### Study design and setting

To assess livestock CCHFV seroprevalence, cross-sectional sampling was conducted among livestock from February to August 2017, and sampling targeted livestock herds that were relatively stable in their location, non-nomadic or actively translocating between districts, and not directly associated with commercial livestock trade networks. Herds were selected to be longitudinally sampled, and this was the first of the longitudinal samples planned to be collected. Sampling was distributed between high-risk districts (based on locations where suspect tick vectors are abundant, especially in northern Uganda) and low-risk districts (mainly consisting of high-altitude areas where suspect tick species are sparse) [15]. Sampling was also performed throughout the Cattle Corridor districts which have a high population of domestic livestock and in districts with international borders. Twenty-eight districts that met at least one of the above criteria were selected for sampling, including Rakai, Nakasongola, Kalangala islands, Nakaseke, Mpigi (Central Uganda), Serere, Mayuge, Bududa, Kamuli, Tororo (Eastern

Uganda), Apac, Arua, Moyo, Karenga, Agago, Lamwo, Moroto, Amudat (Northern Uganda), Isingiro, Ntungamo, Bushenyi, Kamwenge, Kitagwenda, Kiruhura, Buliisa, Kikuube, Bundibugyo, and Kasese (Western Uganda) (Fig 1).

## Sample size calculation and data collection

Livestock serological samples were planned to be tested for IgG antibodies specific to both CCHFV and Rift Valley fever virus (RVFV). Therefore, sample size calculations were conducted individually for each pathogen based on individual effect sizes, estimated seroprevalence, and estimated design effects, and the larger minimum sample size between the two pathogens was selected. Previous estimates of CCHF seroprevalence in domesticated livestock in Uganda and its bordering countries have ranged from 36–76%, therefore we calculated sample size assuming approximately 50% seroprevalence, and aimed to capture an effect size of 5% with 95% confidence[16]. It was necessary to include a design effect given the structured nature of sampling livestock within herds. We used a proportion-to-herd size sampling approach, where we sampled all animals in herds with ≤15 members, and only 25% of animals

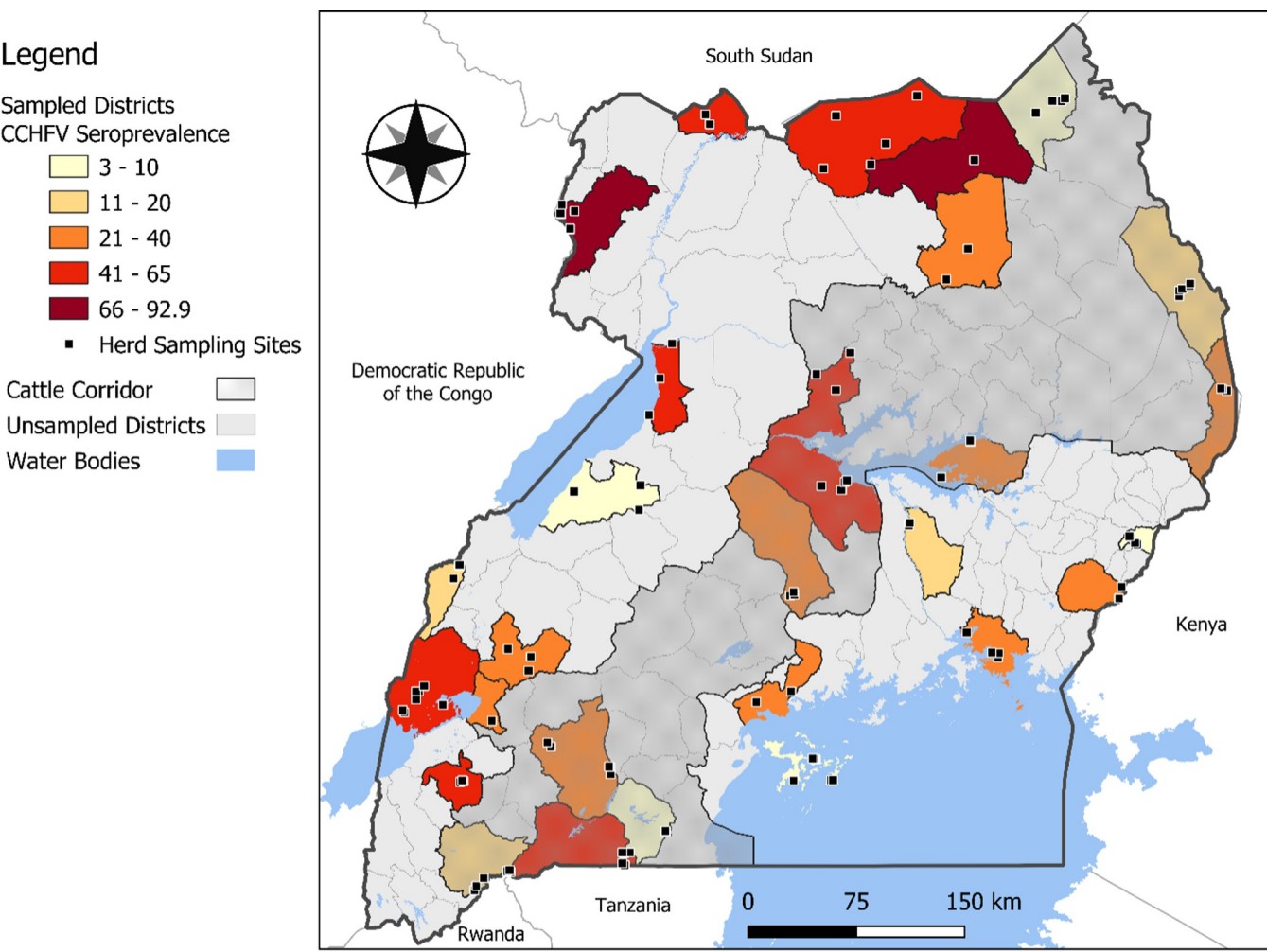

**Fig 1. Sampled districts and their corresponding seroprevalence of Crimean-Congo hemorrhagic fever virus IgG antibodies in cattle, sheep and goats** (Open-source shapefiles for Uganda district boundaries were downloaded from the Humanitarian Data Exchange (Humanitarian Data Exchange, 2020) and water bodies files from the World Bank website (The World Bank, 2022)).

in herds with >15 members. Assuming an average of 15 animals sampled per herd and an intraclass correlation coefficient of 0.2, we calculated a necessary design effect of 3.8 [17]. Therefore, our calculated sample size was 1,460 livestock. The same calculation process was conducted for RVFV using unique seroprevalence and minimum effect size inputs, which resulted in a larger necessary sample size of 2,344 livestock. Assuming an average of 15 animals per herd, we expected to sample 156 herds, distributed evenly throughout the 27 districts selected for sampling. Clusters of herds were purposively selected based on specific criteria, including those with a high tick burden, high animal population and cooperative animal owners, those located in dry and wet areas, and those situated near international borders. Once the clusters were identified, herds with 15 or fewer animals were entirely sampled, while herds with more than 15 animals were sampled using proportional size sampling, where only 25% of the herd was selected for sampling. Individual animals were conveniently chosen and restrained in a crush, and blood samples were collected until the 25% was achieved. During sampling, surveys were conducted with owners of each herd to gather data on animal and herd-specific variables that may be potential predictors of CCHFV seropositivity, including animal species, age (categorized as an infant, juvenile, and adult), sex, breed, management system (grazing pattern), herd size, current and past health status assessed physically by the veterinarian on body condition score and body temperature. We also assessed for abortion or stillbirth history that could be associated with the CCHF virus. Geographic coordinates were also recorded at each sampling site.

## Animal sample collection and laboratory testing

Blood samples were either collected from the jugular vein or the caudal (tail) vein in vacutainer blood collection tubes containing EDTA as a coagulant, immediately aliquoted and stored in liquid nitrogen dewars to maintain the cold chain. Samples were transported to the Uganda Virus Research Institute to be tested for IgG antibodies against CCHFV using an enzyme-linked immunosorbent assay (ELISA) as has been previously described [18, 19]. Briefly, 96-well microtiter plates (Thermo Electron Corporation, Milford, MA, USA) were coated with 100 μL/well of a mouse-derived CCHF capture antibody, prediluted 1:1000 with serum diluent (5% w/v goat skim milk in PBS: pH = 7.4). Plates were incubated overnight at 4˚C, washed 3 times with 250 μL/well wash buffer (PBS containing 0.1% Tween-20v/v), followed by the addition of 100 μL/well of CCHF antigen in the upper half of the plate, and a mock (control) antigen in the lower half of the plate. After 1 hour of incubation at 37˚C, plates were washed with 250 μL/well wash buffer and 33 μL of serum diluent was added to every well. The sample or control sera were diluted at 1:25 in serum diluent and 33 μL of the diluted sample or control sera were added to the plates, with one part added to the antigen half and another to the control half. Aliquots were then subjected to serial 4-fold titrations on the plate, thus making the first and last dilutions in both the antigen or control halves 1 in 100 and 1 in 6,400, respectively. After a 1-hour incubation at 37˚C, plates were washed and 100 μL of rabbit anti-bovine IgG conjugated with horseradish peroxidase (KPL, Gaithersburg, MD) were added to the test wells at a dilution of 1 in 1000 and incubated for 1 hour at 37˚C. The plates were washed and incubated for 30 min at 37˚C with 100 μL/well of 2,2′ -azinobis-(3-ethylbenzothiazoline-6-sulfonic acid) (ABTS) substrate (KPL, Gaithersburg, MD), before being read spectrophotometrically at 490 nm. An adjusted sum optical density (ODSum) for each test and control serum was obtained by adding the differences between the OD values of the control antigen-coated wells from their corresponding CCHF-antigen-coated wells. A positive diagnosis for CCHF IgG in the respective test serum was scored if its ODSum was ≥0.95.

## Data analysis

Data were analyzed using R statistical software [20] for descriptive analysis of animal characteristics and analysis of the relationship between animal characteristics and CCHF seropositivity. In addition to the variables that were gathered from surveys with herd owners during the data collection process, we also ascertained digital elevation data from WorldClim and extracted the elevation values at the coordinates at which animal samples were taken [21]. To quantify the individual relationship between CCHF seropositivity and each variable of interest, a bivariate analysis was first conducted using binomial generalized linear models. For the elevation variable, a cutoff of 1200 meters was used to separate livestock in agro-ecological lowlands from those in agro-ecological midlands and uplands, which tend to be conducive to different vegetation and agricultural patterns that impact tick populations [22]. Following the unadjusted bivariate analysis, a multivariate regression analysis was conducted using a binomial generalized linear mixed model with a random effect for herd sampled, using the R package "lme4" [23]. This multivariate analysis incorporated variables that had <1% missing data, which included animal species, age, sex, breed, and elevation classification. The variance of the herd-level random effect was used to calculate the intraclass correlation coefficient (ICC) to determine the extent to which animals within herds were similar in CCHF seropositivity results. We used the following formula to calculate the ICC:

$$ICC = \sigma/(\sigma + \pi^2/3)$$

Where σ is the variance associated with each herd intercept. A map was created using QGIS 3.28.1 software to visualize the district-level seroprevalence and coordinates of herds that were sampled [24]. Open-source shapefiles for Uganda district boundaries and water bodies were downloaded from the Humanitarian Data Exchange and the World Bank [25, 26].

## Ethical considerations

Approval to do this study was obtained from Uganda Virus Research Institute Research and Ethics Committee and additional approval was obtained from the Uganda National Council of Science and Technology. Animal work associated with this investigation was conducted under CDC Institutional Animal Care and Use Committee protocol number 3098COSMULX and following Uganda national guidelines and performed with officers from the Ministry of Agriculture, Animal Industries and Fisheries. CDC's Human Research Protection Office reviewed and approved the request to allow reliance on a non-CDC IRB for CDC (protocol #7376) per 45 CFR 46.114.

## Results

### Demographics of sampled animals

Enrollment for sampling was higher than expected and a total of 198 herds and 3181 animals were sampled: 1732 cattle (54.4%), 1091 goats (34.3%) and 358 sheep (11.3%). The overall seropositivity of IgG antibodies against CCHFV in all sampled livestock species was 31.4% (999/3181) (Table 1). Seropositivity in cattle was 16.9%, whereas it was 48.7% in goats and 49.2% in sheep. Most animals were adults (70.6%) and females (78.0%), and 71.5% of the sampled animals were bred in Uganda and considered indigenous breeds (Table 1). Most of the animals were healthy at the time of sampling (84.4%) and were kept under communal (45.0%) or paddocking (26.5%) grazing patterns. History of abortion and stillbirth were reported in 21.9% and 10.7% of the female animals, respectively.

**Table 1. Univariate analysis of animal demographics and overall seroprevalence.**

| Variable | Category | Frequency | Percentage |
|---|---|---|---|
| **Species** | Cattle | 1732 | 54.4% |
| | Goats | 1091 | 34.3% |
| | Sheep | 358 | 11.3% |
| **Age** | Infant | 376 | 11.8% |
| | Juvenile | 556 | 17.5% |
| | Adult | 2247 | 70.6% |
| | *Unknown* | *2* | *0.1%* |
| **Sex** | Female | 2482 | 78.0% |
| | Male | 688 | 21.6% |
| | *Unknown* | *11* | *0.4%* |
| **Breed** | Cross-bred | 836 | 26.3% |
| | Exotic | 71 | 2.2% |
| | Indigenous breed | 2274 | 71.5% |
| **CCHF IgG Result** | Negative | 2182 | 68.6% |
| | Positive | 999 | 31.4% |
| **Current Health** | Healthy | 2685 | 84.4% |
| | Unhealthy | 257 | 8.1% |
| | *Unknown* | *239* | *7.5%* |
| **Past Health** | Healthy | 2522 | 79.3% |
| | Unhealthy | 109 | 3.4% |
| | *Unknown* | *550* | *17.3%* |
| **Grazing Pattern** | Paddocking | 842 | 26.5% |
| | Communal | 1430 | 45.0% |
| | Tethering | 170 | 5.3% |
| | Zero Grazing | 113 | 3.5% |
| | *Unknown* | *626* | *19.7%* |
| **Abortion** | No | 683 | 21.5% |
| | Yes | 696 | 21.9% |
| | *Unknown* | *1802* | *56.6%* |
| **Stillbirth** | No | 985 | 31.0% |
| | Yes | 341 | 10.7% |
| | *Unknown* | *1855* | *58.3%* |
| **Elevation** | High | 1403 | 44.1% |
| | Low | 1778 | 55.9% |
| **Region** | Eastern | 525 | 16.5% |
| | Northern | 923 | 29.0% |
| | Central | 663 | 20.8% |
| | Western | 1070 | 33.6% |

## Bivariate analysis of risk factors

An unadjusted bivariate analysis showed that the odds of seropositivity in sheep were 4.8 times the odds among cattle (CI: 3.7–6.1), and the odds of seropositivity among goats was 4.7 times that of cattle (CI: 3.9–5.6) (Table 2). Compared to infants, IgG seroprevalence was significantly higher in juvenile (OR: 2.0; 95% CI: 1.4–2.9) and adult (OR: 3.1; CI: 2.4–4.3) animals and female animals had higher odds of seropositivity compared to male animals (OR: 1.6; 95% CI: 1.3–1.9). The odds of CCHFV seropositivity in the cross and exotic breeds were lower than that among

**Table 2. Unadjusted bivariate analysis of CCHF seropositivity and animal demographics.**

| Variable | Category | CCHF Negative | | CCHF Positive | | |
|---|---|---|---|---|---|---|
| | | n | % | n | % | Unadjusted Odds Ratio (95% CI) |
| **Species** | Cattle | 1440 | 83.1 | 292 | 16.9 | Reference |
| | Goats | 560 | 51.3 | 531 | 48.7 | 4.6 (3.9–5.6) |
| | Sheep | 182 | 50.8 | 176 | 49.2 | 4.8 (3.7–6.1) |
| **Age** | Infant | 320 | 85.1 | 56 | 14.9 | Reference |
| | Juvenile | 411 | 73.9 | 145 | 26.1 | 2.0 (1.4–2.9) |
| | Adult | 1451 | 64.6 | 796 | 35.4 | 3.1 (2.4–4.3) |
| **Sex** | Male | 522 | 75.9 | 166 | 24.1 | Reference |
| | Female | 1650 | 66.5 | 832 | 33.5 | 1.6 (1.3–1.9) |
| **Breed** | Local | 1438 | 63.2 | 836 | 36.8 | Reference |
| | Cross | 675 | 80.7 | 161 | 19.3 | 0.4 (0.3–0.5) |
| | Exotic | 69 | 97.2 | 2 | 2.8 | 0.05 (0.01–0.2) |
| **Grazing Pattern** | Paddocking | 643 | 76.4 | 199 | 23.6 | Reference |
| | Communal | 983 | 68.7 | 447 | 31.3 | 1.5 (1.2–1.8) |
| | Tethering | 83 | 48.9 | 87 | 51.2 | 3.4 (2.4–4.8) |
| | Zero Grazing | 109 | 96.5 | 4 | 3.5 | 0.1 (0.04–0.3) |
| **Abortion** | No | 537 | 78.6 | 146 | 21.4 | Reference |
| | Yes | 467 | 67.1 | 229 | 32.9 | 1.8 (1.4–2.3) |
| **Stillbirth** | No | 766 | 77.8 | 219 | 22.2 | Reference |
| | Yes | 189 | 55.4 | 152 | 44.6 | 2.8 (2.2–3.7) |
| **Elevation** | High | 997 | 71.1 | 406 | 28.9 | Reference |
| | Low | 1185 | 66.6 | 593 | 33.4 | 1.2 (1.1–1.4) |
| **Region** | Eastern | 424 | 80.8 | 101 | 19.2 | Reference |
| | Northern | 551 | 59.7 | 372 | 40.3 | 2.8 (2.2–3.7) |
| | Central | 494 | 74.5 | 169 | 25.5 | 1.4 (1.1–1.9) |
| | Western | 713 | 66.6 | 357 | 33.4 | 2.1 (1.6–2.7) |

local breeds. When considering grazing patterns, we used paddocking as the reference for comparison, although seroprevalence was lower in the zero-grazing group, because the sample size of animals in the zero-grazing group was low. The odds of seropositivity among animals that grazed communally was 1.5 times the odds of animals who were paddocked (CI: 1.2–1.8), and the odds of seropositivity among animals that were tethered was 3.4 times that of animals that were paddocked (CI: 2.4–4.8). Animals under a zero-grazing system had lower odds which were 0.1 times that of paddocked animals (CI: 0.04–0.3). The odds of seropositivity in animals with a history of abortion or stillbirth were higher than that among animals without a history of abortion or stillbirth. Considering geographical region, the odds of CCHFV IgG seropositivity was significantly higher in the northern, western and central districts when compared to the eastern districts, and animals sampled at low elevation had higher odds of seropositivity compared to animals sampled at higher elevations (OR: 1.3; 95% CI: 1.1–1.5).

## Multivariate analysis of risk factors

In an adjusted binomial generalized linear mixed regression model, the association between animal species and CCHF seropositivity remained statistically significant, where the odds of seropositivity among goats and sheep were 4.4 (CI: 3.3–6.0) and 3.7 (CI: 2.5–5.7) times the odds among cattle, respectively. Likewise, age group and sex also remained significantly associated with CCHF seropositivity, where the odds of seropositivity among adult and juvenile

livestock were 2.7 (CI: 1.9–3.9) and 1.7 (CI: 1.1–2.6) times the odds among infants, respectively, and the odds of seropositivity in females was 1.3 (CI: 1.0–1.7) times the odds in males. Holding all other variables in the model constant, the associations between CCHF seropositivity and animal breed and elevation were not statistically significant based on a 95% confidence limit (Table 3).

Considering the random effect for animal herds, we calculated an ICC of 0.4, indicating high clustering of CCHF seropositivity within herds.

## Discussion

Given the emergence of CCHF in humans in Uganda in 2013, there was a need to investigate the distribution of CCHFV seroprevalence in livestock across the country, which acts as the primary source of infection for humans. This study represents the most comprehensive nationwide CCHF IgG serosurvey in livestock ever performed in Uganda, where serological samples were collected from herds of cattle, sheep, and goats to estimate the proportion of livestock exposed to CCHFV and identify animal characteristics associated with the odds of CCHF seropositivity. We collected blood samples from 3181 cattle, sheep, and goats from 198 herds in 27 districts throughout Uganda, which represented the varying geographic and ecological regions of the country. Overall, IgG antibodies against CCHFV were present in 31.4% (999/3181) of livestock. Seroprevalence was higher among sheep (49.2%) and goats (48.7%) compared to cattle (16.9%). In a multivariate binomial mixed effects model, we found that animal species, age group, and sex were significantly associated with CCHFV seropositivity. We found a herd-level ICC of 0.4, suggesting high clustering of CCHFV seropositivity within herds.

Our findings of livestock seroprevalence were similar to results found in previous serosurveys conducted in Uganda and other African countries. A recent study from Uganda reported regional CCHFV seropositivity of 15% in cattle species using the same diagnostic assay [18] as our study and is similar to our study findings of 16.9% seropositivity in sampled cattle. However, Balinandi et al. (2021) performed serology tests on cattle samples in three of the studied districts using the ID screen CCHF double antigen multi-species (IDVet), a commercial CCHF serological testing kit. Their findings revealed a seropositivity rate of 75% in cattle. This suggests that the selection of an ELISA assay could potentially result in variations in seropositivity rates [18, 27]. No other previous studies have tested CCHF IgG antibodies in small ruminants (goat and sheep) in Uganda, and the seropositivity of approximately 49% was higher

**Table 3. Multivariate logistic regression model for CCHF seropositivity in animals.**

| Variable | Category | Odds Ratio (95% CI) | P-value |
|---|---|---|---|
| **Species** | Cattle | *Reference* | - |
| | Goat | 4.4 (3.3–6.0) | <0.001 |
| | Sheep | 3.7 (2.5–5.7) | <0.001 |
| **Age** | Infant | *Reference* | - |
| | Juvenile | 1.7 (1.1–2.6) | 0.01 |
| | Adult | 2.7 (1.9–3.9) | <0.001 |
| **Sex** | Male | *Reference* | - |
| | Female | 1.3 (1.0–1.7) | 0.04 |
| **Breed** | Local | *Reference* | - |
| | Cross | 0.7 (0.5–1.1) | 0.14 |
| | Exotic | 0.3 (0.03–1.0) | 0.09 |
| **Elevation** | High | *Reference* | - |
| | Low | 1.5 (0.8–2.7) | 0.20 |

than expected [3]. In a meta-analysis of seroprevalence of CCHFV in livestock, Hasan *et al* (2019) also reported similar results where the seroprevalence in cattle (18%) was lower compared to sheep (24%) and goats (29%), resulting in an overall seroprevalence of 24.6% [3]. A study in northwestern Senegal found a seroprevalence of 32%, although seroprevalence in sheep was 22% and in goats was 9%, which were also lower than what we found in these species [28]. This could be attributed to the difference in sensitivity and specificity of test kits used to measure the seropositivity of CCHF in domestic animals. However, the epidemiology of CCHF may also differ significantly from one region to another influenced by tick vector dynamics, tick vector control methods and climate. Studies in Kenya have found CCHF seroprevalence of 32% in domesticated ruminants, and as high as 75% in buffalo [29, 30]. This could explain the higher seropositivity we found in livestock that grazed near Uganda national parks such as those in northeastern Uganda in the Kidepo Valley National Park ecosystem and around Lake Mburo National Park where the buffalo interact with livestock and could potentially facilitate transmission to livestock and subsequent spillover into humans.

One of the most notable findings from our study was the stark difference in seroprevalence between small ruminants (sheep and goats) and cattle. Tick control among livestock in Uganda often does not include small ruminants, which are not usually treated with acaricides as frequently as cattle. During our sample collection process, we found a higher tick burden in sheep and goats compared to cattle, suggesting less tick control efforts by livestock owners for small ruminants and explaining the elevated seropositivity in these species. Different species of ticks are hypothesized to be potential transmitters of ticks in Uganda and studies are ongoing to determine the exact tick vector in Uganda [11, 31, 32].

Similarly, we found that CCHF seroprevalence was higher in local breeds compared to exotic breeds, which could be explained by the fact that exotic breeds are prioritized by farmers for tick control as they are more susceptible to tick-borne diseases [33]. Recent CCHFV incidence among humans has been linked to close contact with goats and sheep [14]. Future efforts to mitigate the risk of spillover of CCHFV from livestock to humans may be most beneficial if focused on small ruminants.

Seropositivity was higher among animals that were communally grazed or tethered compared to those that were paddocked, however, this relationship could not be tested in an adjusted multivariate model due to a large proportion of missing data. CCHFV seropositivity was slightly larger at lower elevation (33.4%) compared to high elevations (28.9%), but after adjusting for species, age, sex, and breed we did not find that the odds of seropositivity was significantly larger among animals at low elevations. Seroprevalence was also higher in animals with a reported history of stillbirth and abortion, but this relationship also could not be tested in an adjusted model due to large proportions of missing data.

Nevertheless, since animals infected with the CCHF virus are not typically symptomatic, it is crucial to delve deeper into the effects of CCHFV infection on animal production, specifically in terms of potential reductions in herd size and milk production. However, it is important to consider that the interpretation and correlation between CCHFV seropositivity and stillbirth or abortion may be influenced by confounding factors. This is because the same animal populations in Uganda are susceptible to other diseases, such as brucellosis and Rift Valley fever virus, which are known to cause abortions.

Seropositivity was also higher in the north and northeastern regions of Uganda, which is known to have warmer average temperatures compared to the western and central parts of Uganda. This seems to be typical ecology for the survival of the tick vectors as they are known to survive in warmer climates as opposed to the colder climate, however, a formal spatial analysis should be conducted to make inferences about the spatial distribution of CCHF throughout the country.

We found that the odds of CCHF seropositivity were significantly higher among older livestock compared to younger livestock in both the bivariate analysis and the adjusted multivariate analysis. Increased seroprevalence among older animals compared to younger animals was expected given that the expected duration of IgG antibodies is longer than the typical life expectancy of most domesticated ungulates, and younger animals have fewer opportunities to be exposed compared to older animals. Female animals also had higher odds of seroprevalence compared to males. This may be because female cattle and goats tend to have longer lives than males given their use for milk production.

We accounted for the hierarchical nature of our herd sampling data by using a mixed effects model, wherein we found an ICC for livestock herds of 0.4, suggesting a high degree of relatedness in CCHF seropositivity between animals in the same herd. Compared to other studies of CCHF, which have found a within-herd correlation of 0.29 in Cameroon [34], 0.3 in Zambia [35], and 0.19 in South Africa [36] our findings suggest a higher within-herd correlation of CCHFV seropositivity in Uganda than has been seen in other countries. Uganda's relatively high ICC suggests that CCHF seroprevalence is more likely to vary between herds, which should be accounted for when conducting future sampling efforts and explored further using formal spatial analysis methods to predict the distribution of CCHFV across the country to account for the degree of correlation between herds as a function of distance.

Purposive sampling could be one of the limitations of this study especially since sampling was biased towards what we considered high-risk areas. Examples are places that reported human outbreaks and ecological zones that favour tick vector survival. There is a need to design a follow-up study that is random without bias towards regions where the disease is expected. Also, the assay used is an in-house assay that tends to underestimate the prevalence of CCHF as demonstrated by Balinandi et al, 2021 [18].

In conclusion, we have demonstrated a high prevalence of CCHFV IgG antibodies in Ugandan livestock, ranging from 16.9% in cattle to 49.2% in sheep and 48.7% in goats, resulting in an overall domesticated livestock seroprevalence of 31.4%. Spillover into the human population could potentially be reduced by targeting surveillance and transmission mitigation efforts towards higher-risk demographics of livestock, such as sheep and goats. While livestock plays an important role in CCHFV spillover into humans, ticks also play an important role in the lifecycle and transmission dynamics of CCHFV and additional studies investigating the influence that infected tick populations have on livestock infections and human spillover. This will help to further refine CCHFV transmission mitigation efforts and tick control measures and thus reduce the burden of tick-borne diseases, particularly CCHFV. Additionally, data collected from this study will be used for additional analysis looking at ecological and environmental variables that are predictive of CCHFV to generate a map of estimated CCHF seroprevalence in unsampled locations across Uganda. Ultimately, all results and analysis from this study will be used to target specific regions for enhanced human and livestock surveillance and help guide the introduction of new rapid diagnostic diagnostics for more rapid case detection.

## Acknowledgments

We thank the district veterinary officers, animal husbandry officers, farmers, herdsmen and community leaders for their great help in conducting this research in their communities and their areas of jurisdiction. Special thanks also go to the laboratory staff of different health facilities across the country who gave us space to process the samples in the field. Special thanks go to Sam Twongyeirwe, Apolo Bogere, Eriya Sembusi, Sinani Kigozi, Amia Winnie, Kilama Kamugisa and Gloria Akurut for their great help during fieldwork.

**Disclaimer:** The findings and conclusions in this report are those of the authors and do not necessarily represent the official position of the Centers for Disease Control and Prevention or any institutions with which the authors are affiliated.

## Author Contributions

**Conceptualization:** Luke Nyakarahuka, Julius J. Lutwama, Stuart T. Nichol, Stephen K. Balinandi, Trevor R. Shoemaker.

**Data curation:** Luke Nyakarahuka, Jackson Kyondo, Carson Telford, Amy Whitesell, Sophia Mulei, Jimmy Baluku, Deborah L. Cannon, Trevor R. Shoemaker.

**Formal analysis:** Luke Nyakarahuka, Jackson Kyondo, Carson Telford, Amy Whitesell, Alex Tumusiime, Sophia Mulei, Caitlin M. Cossaboom, Deborah L. Cannon, Stephen K. Balinandi, John D. Klena.

**Funding acquisition:** Joel M. Montgomery, Julius J. Lutwama, Stuart T. Nichol, John D. Klena, Trevor R. Shoemaker.

**Investigation:** Luke Nyakarahuka, Jackson Kyondo, Alex Tumusiime, Sophia Mulei, Jimmy Baluku, Stephen K. Balinandi, Trevor R. Shoemaker.

**Methodology:** Luke Nyakarahuka, Jackson Kyondo, Carson Telford, Amy Whitesell, Alex Tumusiime, Sophia Mulei, Jimmy Baluku, Caitlin M. Cossaboom, Deborah L. Cannon, Joel M. Montgomery, Julius J. Lutwama, Stuart T. Nichol, Stephen K. Balinandi, John D. Klena, Trevor R. Shoemaker.

**Project administration:** Luke Nyakarahuka, Caitlin M. Cossaboom, Joel M. Montgomery, Julius J. Lutwama, Stuart T. Nichol, Stephen K. Balinandi, John D. Klena, Trevor R. Shoemaker.

**Resources:** Luke Nyakarahuka, Deborah L. Cannon, Joel M. Montgomery, Julius J. Lutwama, Stuart T. Nichol, Stephen K. Balinandi, John D. Klena, Trevor R. Shoemaker.

**Software:** Luke Nyakarahuka, Carson Telford, Amy Whitesell, Jimmy Baluku.

**Supervision:** Luke Nyakarahuka, Jackson Kyondo, Caitlin M. Cossaboom, Deborah L. Cannon, Joel M. Montgomery, Julius J. Lutwama, Stuart T. Nichol, Stephen K. Balinandi, John D. Klena, Trevor R. Shoemaker.

**Validation:** Luke Nyakarahuka, Jackson Kyondo, Carson Telford, Amy Whitesell, Alex Tumusiime, Sophia Mulei, Jimmy Baluku, Caitlin M. Cossaboom, Deborah L. Cannon, Joel M. Montgomery, Julius J. Lutwama, Stuart T. Nichol, Stephen K. Balinandi, John D. Klena, Trevor R. Shoemaker.

**Visualization:** Luke Nyakarahuka, Carson Telford, Amy Whitesell, Jimmy Baluku, Trevor R. Shoemaker.

**Writing – original draft:** Luke Nyakarahuka, Trevor R. Shoemaker.

**Writing – review & editing:** Luke Nyakarahuka, Jackson Kyondo, Carson Telford, Amy Whitesell, Alex Tumusiime, Sophia Mulei, Jimmy Baluku, Caitlin M. Cossaboom, Deborah L. Cannon, Joel M. Montgomery, Julius J. Lutwama, Stuart T. Nichol, Stephen K. Balinandi, John D. Klena, Trevor R. Shoemaker.

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
