## [Decision Letter · Decision Letter 0]

9 Sep 2022

PONE-D-22-22276Seroepidemiological Investigation of Crimean Congo Hemorrhagic fever virus in Livestock in Uganda, 2017PLOS ONE

Dear Dr. Luke Nvakarahuka, 

Thank you for submitting your manuscript to PLOS ONE. After careful consideration, we feel that it has merit but does not fully meet PLOS ONE’s publication criteria as it currently stands. Therefore, we invite you to submit a revised version of the manuscript that addresses the points raised during the review process.

ACADEMIC EDITOR: The manuscript needs a revision in the writing process and language editing. The reviewers comments should be replied clearly.  

We look forward to receiving your revised manuscript.

Kind regards,

Shawky M Aboelhadid, PhD

Academic Editor

PLOS ONE

Reviewers' comments:

Reviewer's Responses to Questions

**Comments to the Author**

1. Is the manuscript technically sound, and do the data support the conclusions?

Reviewer #1: Partly

2. Has the statistical analysis been performed appropriately and rigorously? 

Reviewer #1: I Don't Know

3. Have the authors made all data underlying the findings in their manuscript fully available?

Reviewer #1: No

4. Is the manuscript presented in an intelligible fashion and written in standard English?

Reviewer #1: No

5. Review Comments to the Author

Reviewer #1: The study shows the results of a comprehensive serological survey looking at the seroprevalence of CCHFv in livestock in Uganda. The study included 3 different species known to be amplifier hosts for the virus and the risk factors associated with seropositivity are explored. CCHF is a disease of public health importance, and in Uganda the recent identification of clinical cases supports the need of further investigations aimed to understand its epidemiology and project the risk for humans, especially at-risk populations. It was great to see that the whole country was represented in the survey, and the large sample size is a positive aspect. While I recommend revisions, I am aware of the potential of this manuscript once these aspects are addressed.

Comments

Major

Improve its clarity and the flow of ideas of the manuscript. At the moment, it is not easy to read the manuscript and some changes are required to make the writing flow so that the story and the message are clear.

From an epidemiological perspective, there seems to be an underlying well-thought study design that is crucial for the conclusions made for the country. However, many important details were not included in the text leaving many questions unanswered for potential readers. The authors indicate that the sample size is one of the greatest strengths but without further information to understand the design and the context the large number of animals sampled loses relevance.

The results could be more informative. Considering the scope of the analysis in a study conducted in several districts and including 3 species, many more details that are fundamental to understand the burden of the disease in the country and the implications of the findings.

The discussion requires a more critical point of view. Currently, it is focused in comparing the results of the survey with the results from previous studies including systematic reviews but considering the complexity of CCHFv with a clear ecological component (vector, reservoirs, transmission to humans), the discussion should be able to critically explain this results in relation to the design, the implications and the limitations of the analysis conducted.

The authors are referred to The STROBE Statement – Checklist of items that should be included in reports of cross-sectional studies to improve the manuscript.

Minor

Requires more attention to little details including punctuation, use of language, and percentages presented.

Notes

- Keywords should be indexed terms.

- More details of the funders as indicated in the guidance.

1. Abstract

“Adult animals 36 represented 70.6% and 78% of the sampled animals were females whereas local breeds represented 71.5%” (Line 35-37). Not clear what the two percentages mean as it seems to be talking only about adult animals, so I would expect only one percentage. Please review this idea and re-write for clarity.

Worried about making conclusions about spatial trends without formally testing (Line 39-40).

For clarity, invert the risk across species and start by indicating that sheep and goats had a higher risk, otherwise, it is not immediately clear how cattle had lower odds when only the OR for sheep and goats is presented (Line 40-41)

Line 44: Missing p for p value of p>0.01. A little misleading that some of the differences are expressed as OR with confidence intervals and other with p-values. I suggest standardising for the abstract.

Would be careful to say that the study shows that CCHFv is endemic in Uganda based on this (Line 50-51). Maybe say is actively circulating? What are the criteria to declare a disease endemic? Maybe as it is the first serological survey of this type, more evidence is required before making such a conclusion.

For the abstract, I suggest focussing on the results of the multivariable model in more detail. Presenting both univariate and multivariable results can be misleading.

2. Background

Complete some references (e.g. Line 56) and review punctuation (e.g. Line 63). Along the text there are many statements including the reports of the cases in the districts, the percentage of animals positive in Wakiso and Kiboga, etc that would need a reference to the report or at least the source which I assume is the Ugandan Virus Research Institute.

CCHFv is transmitted directly and indirectly, but for animals in the absence of clinical signs it is more frequently tick-mediated. Indicating that is animal-to-animal gives the wrong impression, in my opinion (Line 59).

While the information contained in the introduction is relevant, it lacks writing flow, so it is not an engaging introduction to read. I would suggest splitting into small paragraphs and improving the connections between sentences and paragraphs for a more enjoyable read.

When I read the objective, I think about the seropositivity but it is also clear that you’re looking to explore the risk factors associated to seropositivity and to assess the links to reproductive problems. Would be worth mentioning here for clarity and to manage expectations of readers.

Would be interesting to mention which tick vectors are involved. This would give some information to the readers in relation to why you’re thinking CCHFv might be found in Uganda, even in districts that have not reported cases. Is it likely that the ticks that transmit it are there?

3. Methods

Study design: All the information is there but once again I find that the writing and flow of ideas needs to be improved. For instance: Start with study locations and how/why these were selected and later move on to describe the types of herds included. Additional details of the classification of the districts into high and low risk is important, especially because you have not mentioned before any details of the possible tick vectors implicated in transmission. Of the list, which are the high-risk districts, which are low risk districts and which ticks are found there. I suggest including a map to visually present this information, as this part of the design is crucial for the conclusions made later on.

Sample size calculation and data collection: Reference the software used for calculating the sample size. Also, all the parameters including error should be disclosed so this calculation can be replicated (Line 107-108). Once the sample size is calculated how was it distributed across districts? What is the selection criteria for the animals (inclusion and exclusion) including age, sex, production system and how were the species distributed? Was it a random or convenience sample? Was it stratified in any way, for instance the production system (transhumant vs non-nomadic)? How were herds selected? There needs to be clarity of the population selected to be able to interpret the results of the study appropriately.

Data analysis: More information of the procedure used from analysis. You started with a bivariate analysis, but no detailed tests are included. Same for the multivariable model, which model (I assume it is a logistic regression), however, more details are required as to how variables are selected in the model, if the structured nature of the data (herds within districts) was considered and how you selected and assessed the final model. If R was used for data analysis, the packages should be included and referenced. In addition, there seems to be an additional hypothesis being tested here related to the reproductive history. I am aware that this needs further evidence but at the moment it is presented without contexts as to why you’re looking for this association if the initial goal and justification for the study was mainly to identify which areas might be at risk using livestock as sentinels for CCHFv. If this part is kept, it needs to be additional information to introduce why this is interesting and then the gap that this is filling. Also details on how many animals have information of this and how we can trust it given that it is incomplete. How did you choose the cut off point for elevation? Does it make sense to think about it like this considering possible ideal habitats for the ticks?

Ethics statement: Not clear, partially because there are many acronyms that should be defined (e.g., CFR, IBR, UVRI). I suggest to re-write this part and improve the flow of information presented for clarity.

4. Results

Table 1: Does not present the overall seroprevalence but the description of the study population. It would be good to be clear on the cut-off points or definitions for some of the variables. For example, criteria to classify an animal as an infant, ‘medium’, and adult. Also, what was accepted as healthy vs unhealthy for current and past health (eligibility criteria), abortion and still birth (definition and how did you ask this question (timeframe considered)). Most of these aspects should be better described in the methods so the reader is clear by the time the information is presented here. Were there differences in these categories between species? It would be interesting to know how the population was distributed in relation to the location (as suggested before) and also, in relation to the main features.

Bivariate analysis of risk factors: Same comment as before for the way to present the comparative risk across species (Lines 186 – 188).

Please standardise the results p-values or OR. It is more informative to present OR for all as it indicates the magnitude of the risk.

Table 2: Some of the choices of reference for the comparison are odd. Normally, the reference category is the one that is believed to be at a lower risk and in some of these I don’t understand how this was chosen. For instance, animals zero grazed might be at lower risk than paddocking and communal because the later roam free and are more exposed to ticks. The selection of the reference category needs further thought/justification/discussion.

Table 3: Same comments as before, consider how the variable that you are evaluating influences the risk and then this makes more sense when analysing and interpreting in the discussion.

The model that you chose to present is simple considering the structured population (does not include random effects). However, I wonder about the differences between the non-nomadic vs transhumant districts, the high and low-risk districts that you described in the methods, and also the locations that are in the border vs the ones that are not. Lastly, were there fundamental differences across locations or the systems in the districts that explain these differences? what about the different species? Does anything change when you analyse their risk separately?

The results could have been a little bit more informative, based on the data you collected and the even if the initial aim is only mapping, describing, and evaluating individual risk, my perception is that the study falls short on addressing fundamental aspects related to the epidemiology of CCHF.

The district names in the figure are not readable.

5. Discussion:

Line 227: “We designed a study 227 to estimate the burden of the disease in livestock to come up with risk-based health surveillance models for RVF” – Please review should say CCHF. Also, if this is a risk-based health surveillance as indicated here, more details should be included about this in the methods and in the results to support this claim and the results obtained.

Line 239-234: Maybe commercial essays overestimate but there is no way to be sure. Reasons for variations in seroprevalence are multiple. Unless to you compared and you’re performing quality control of the results by running them in duplicate or any other strategy, not sure if the performance of the diagnostic test is the only possible explanation for this difference. What about real differences? Timing? Population?

It would have been interesting to have a discussion in relation to the findings of districts that have previously reported the cases, considering that some of these were sampled as part of this survey (e.g. Agago).

Need to discuss the limitations/possible biases of the study/design and how this affects the conclusions. As well as further perspectives.

6. PLOS authors have the option to publish the peer review history of their article (what does this mean?). If published, this will include your full peer review and any attached files.

Reviewer #1: No

---

## [Author Response · Author response to Decision Letter 0]

16 Mar 2023

Response to reviewers. 

Improve its clarity and the flow of ideas of the manuscript. At the moment, it is not easy to read the manuscript and some changes are required to make the writing flow so that the story and the message are clear.

The manuscript has been edited for flow and the message is clear now. 

From an epidemiological perspective, there seems to be an underlying well-thought study design that is crucial for the conclusions made for the country. However, many important details were not included in the text leaving many questions unanswered for potential readers. The authors indicate that the sample size is one of the greatest strengths but without further information to understand the design and the context the large number of animals sampled loses relevance.

We have greatly improved this component on sampling and added it to the manuscript and thus; 

Livestock serological samples were planned to be tested for IgG antibodies specific to both CCHFV and Rift Valley fever virus (RVFV). Therefore, sample size calculations were conducted individually for each pathogen based on individual effect sizes, estimated seroprevalence, and estimated design effects, and the larger minimum sample size between the two pathogens was selected. Previous estimates of CCHF seroprevalence in domesticated livestock in Uganda and its bordering countries have ranged from 36-76%, therefore we calculated sample size assuming approximately 50% seroprevalence, and aimed to capture an effect size of 5% with 95% confidence (Spengler et al., 2016). It was necessary to include a design effect given the structured nature of sampling livestock within herds. We used a proportion-to-herd size sampling approach, where we sampled all animals in herds with ≤15 members, and only 25% of animals in herds with >15 members. Assuming an average of 15 animals sampled per herd and an intraclass correlation coefficient of 0.2, we calculated a necessary design effect of 3.8 (Otte & Gumm, 1997). Therefore, our calculated sample size was 1,460 livestock. The same calculation process was conducted for RVFV using unique seroprevalence and minimum effect size inputs, which resulted in a larger necessary sample size of 2,344 livestock. Assuming an average of 15 animals per herd, we expected to sample 156 herds, distributed evenly throughout the 27 districts selected for sampling. During sampling, surveys were conducted with owners of each herd to gather data on animal and herd-specific variables that may be potential predictors of CCHFV seropositivity, including animal species, age, sex, breed, management system (grazing pattern), current and past health status, herd size, and health history. Geographic coordinates were also recorded at each sampling site. 

The results could be more informative. Considering the scope of the analysis in a study conducted in several districts and including 3 species, many more details are fundamental to understanding the burden of the disease in the country and the implications of the findings.

Thank you for this comment. We have analyzed the results and interpreted them better. 

The discussion requires a more critical point of view. Currently, it is focused on comparing the results of the survey with the results from previous studies including systematic reviews but considering the complexity of CCHFv with a clear ecological component (vector, reservoirs, transmission to humans), the discussion should be able to critically explain this results in relation to the design, the implications and the limitations of the analysis conducted.

We have looked into this recommendation and discussed our results accordingly. 

The authors are referred to The STROBE Statement – Checklist of items that should be included in reports of cross-sectional studies to improve the manuscript.

We have checked the manuscript against The STROBE Statement to improve its reporting quality and standard and made sure all components of reporting cross-section studies are included in the manuscript. 

Minor

Requires more attention to little details including punctuation, use of language, and percentages presented.

 We checked this and made all the grammatical errors that we could detect 

Notes

- Keywords should be indexed terms.

- More details of the funders as indicated in the guidance.

This has been edited accordingly. 

1. Abstract

“Adult animals 36 represented 70.6% and 78% of the sampled animals were females whereas local breeds represented 71.5%” (Line 35-37). Not clear what the two percentages mean as it seems to be talking only about adult animals, so I would expect only one percentage. Please review this idea and re-write for clarity.

We edited this section in the abstract and removed the confusing statement 

Worried about making conclusions about spatial trends without formally testing (Line 39-40).

Will indicated that the spatial trends are not obvious, and we are not making any conclusions. 

For clarity, invert the risk across species and start by indicating that sheep and goats had a higher risk, otherwise, it is not immediately clear how cattle had lower odds when only the OR for sheep and goats is presented (Line 40-41)

This has been fixed in the result section and the abstract edited accordingly. 

Line 44: Missing p for p value of p>0.01. A little misleading that some of the differences are expressed as OR with confidence intervals and other with p-values. I suggest standardising for the abstract.

We have now edited the whole manuscript and replaced the p-values with 95% confidence intervals of Odds Ratios since they give more meaning compared to p-values. 

Would be careful to say that the study shows that CCHFv is endemic in Uganda based on this (Line 50-51). Maybe say is actively circulating? What are the criteria to declare a disease endemic? Maybe as it is the first serological survey of this type, more evidence is required before making such a conclusion.

Agreed. This has been edited accordingly as advised. 

For the abstract, I suggest focussing on the results of the multivariable model in more detail. Presenting both univariate and multivariable results can be misleading.

We edited the abstract extensively and focused on the results of the multivariable model, leaving the univariate and other details for the result section. 

2. Background

Complete some references (e.g. Line 56) and review punctuation (e.g. Line 63). Along the text there are many statements including the reports of the cases in the districts, the percentage of animals positive in Wakiso and Kiboga, etc that would need a reference to the report or at least the source which I assume is the Ugandan Virus Research Institute.

We have added the references in these sections. 

CCHFv is transmitted directly and indirectly, but for animals in the absence of clinical signs, it is more frequently tick-mediated. Indicating that is animal-to-animal gives the wrong impression, in my opinion (Line 59).

Yes, we agree that this was confusing and we have edited it out. 

While the information contained in the introduction is relevant, it lacks writing flow, so it is not an engaging introduction to read. I would suggest splitting into small paragraphs and improving the connections between sentences and paragraphs for a more enjoyable read.

We have edited the introduction to improve flow and corrected for grammatical errors to improve readability. 

When I read the objective, I think about the seropositivity but it is also clear that you’re looking to explore the risk factors associated to seropositivity and to assess the links to reproductive problems. Would be worth mentioning here for clarity and to manage expectations of readers.

 This has been modified to reflect the results and the finding presented in the manuscript. 

Would be interesting to mention which tick vectors are involved. This would give some information to the readers in relation to why you’re thinking CCHFv might be found in Uganda, even in districts that have not reported cases. Is it likely that the ticks that transmit it are there?

 We mention the ticks that have been described to be potential vectors for CCHF in Uganda in the manuscript, mainly Rhipicephalus and Boophilus species which are abundant in Uganda and have added the references for some of these studies in the discussion. 

3. Methods

Study design: All the information is there but once again I find that the writing and flow of ideas needs to be improved. For instance: Start with study locations and how/why these were selected and later move on to describe the types of herds included. Additional details of the classification of the districts into high and low risk is important, especially because you have not mentioned before any details of the possible tick vectors implicated in transmission. Of the list, which are the high-risk districts, which are low risk districts and which ticks are found there. I suggest including a map to visually present this information, as this part of the design is crucial for the conclusions made later on.

We have improved this component and Figure 1 shows the sampled districts and their locations. We do not have a clear distribution of tick species in Uganda as studies are still ongoing on which species is predominant in which region. 

Sample size calculation and data collection: Reference the software used for calculating the sample size. Also, all the parameters including errors should be disclosed so this calculation can be replicated (Line 107-108). Once the sample size is calculated how was it distributed across districts? What is the selection criteria for the animals (inclusion and exclusion) including age, sex, production system and how were the species distributed? Was it a random or convenience sample? Was it stratified in any way, for instance the production system (transhumant vs non-nomadic)? How were herds selected? There needs to be clarity of the population selected to be able to interpret the results of the study appropriately.

Thank you for these observations, we have edited this section and included the information requested. 

Data analysis: More information of the procedure used from the analysis. You started with a bivariate analysis, but no detailed tests are included. Same for the multivariable model, which model (I assume it is a logistic regression), however, more details are required as to how variables are selected in the model, if the structured nature of the data (herds within districts) was considered and how you selected and assessed the final model. If R was used for data analysis, the packages should be included and referenced. In addition, there seems to be an additional hypothesis being tested here related to the reproductive history. I am aware that this needs further evidence but at the moment it is presented without contexts as to why you’re looking for this association if the initial goal and justification for the study was mainly to identify which areas might be at risk using livestock as sentinels for CCHFv. If this part is kept, it needs to be additional information to introduce why this is interesting and then the gap that this is filling. Also details on how many animals have information of this and how we can trust it given that it is incomplete. How did you choose the cut off point for elevation? Does it make sense to think about it like this considering possible ideal habitats for the ticks?

Thank you for these comments. We have incorporated these comments and improved the data analysis section as advised by the reviewer. 

Ethics statement: Not clear, partially because there are many acronyms that should be defined (e.g., CFR, IBR, UVRI). I suggest to re-write this part and improve the flow of information presented for clarity.

This has been improved and acronyms defined. 

4. Results

Table 1: Does not present the overall seroprevalence but the description of the study population. It would be good to be clear on the cut-off points or definitions for some of the variables. For example, criteria to classify an animal as an infant, ‘medium’, and adult. Also, what was accepted as healthy vs unhealthy for current and past health (eligibility criteria), abortion and still birth (definition and how did you ask this question (timeframe considered)). Most of these aspects should be better described in the methods so the reader is clear by the time the information is presented here. Were there differences in these categories between species? It would be interesting to know how the population was distributed in relation to the location (as suggested before) and also, in relation to the main features.

We improved the explanation of this in the methods section. Since we measured IgG antibodies that are expected to last for long periods, we were not strict in terms of timelines for health history or abortion history. 

Bivariate analysis of risk factors: Same comment as before for the way to present the comparative risk across species (Lines 186 – 188).

This has been edited to read well across species comparison. 

Please standardise the results p-values or OR. It is more informative to present OR for all as it indicates the magnitude of the risk.

Yes, we have used majorly Odds Ratios and their 95% Confidence intervals throughout the manuscript. 

Table 2: Some of the choices of reference for the comparison are odd. Normally, the reference category is the one that is believed to be at a lower risk and in some of these I don’t understand how this was chosen. For instance, animals zero grazed might be at lower risk than paddocking and communal because the later roam free and are more exposed to ticks. The selection of the reference category needs further thought/justification/discussion.

We evaluated each variable and agreed on which would be the best reference point depending on sample size or risk level. For example, when considering grazing patterns, we used paddocking as the reference for comparison, although seroprevalence was lower in the zero-grazing group because the sample size of animals in the zero-grazing group was low. This has been added to the manuscript. 

Table 3: Same comments as before, consider how the variable that you are evaluating influences the risk and then this makes more sense when analysing and interpreting in the discussion.

We have edited Table 3 and considered the recommendation. 

The model that you chose to present is simple considering the structured population (does not include random effects). However, I wonder about the differences between the non-nomadic vs transhumant districts, the high and low-risk districts that you described in the methods, and also the locations that are in the border vs the ones that are not. Lastly, were there fundamental differences across locations or the systems in the districts that explain these differences? what about the different species? Does anything change when you analyse their risk separately?

Following the unadjusted bivariate analysis, a multivariate regression analysis was conducted using a binomial generalized linear mixed model with a random effect for herd sampled, using the R package “lme4” (Bates, et al.,(Bates et al., 2015). This multivariate analysis incorporated variables that had <1% missing data, which included animal species, age, sex, breed, and elevation classification. The variance of the herd-level random effect was used to calculate the intraclass correlation coefficient (ICC) to determine the extent to which animals within herds were similar in CCHF seropositivity results. We used the following formula to calculate the ICC:

ICC = σ/(σ+π2/3)

Where σ is the variance associated with each herd intercept. We have added this in the manuscript in the methods section. 

The results could have been a little bit more informative, based on the data you collected and the even if the initial aim is only mapping, describing, and evaluating individual risk, my perception is that the study falls short of addressing fundamental aspects related to the epidemiology of CCHF.

 This manuscript is the first of the kind with a big national wide coverage in terms of sampling and providing critical data for the epidemiology of CCHF in animals. 

The district names in the figure are not readable.

The figure has been edited to make it more readable. 

5. Discussion:

Line 227: “We designed a study 227 to estimate the burden of the disease in livestock to come up with risk-based health surveillance models for RVF” – Please review should say CCHF. Also, if this is a risk-based health surveillance as indicated here, more details should be included about this in the methods and in the results to support this claim and the results obtained.

Thank you for identifying this error, we have extensively edited the manuscript and removed such errors. 

Line 239-234: Maybe commercial essays overestimate but there is no way to be sure. The reasons for variations in seroprevalence are multiple. Unless to you compared and you’re performing quality control of the results by running them in duplicate or any other strategy, not sure if the performance of the diagnostic test is the only possible explanation for this difference. What about real differences? Timing? Population?

 Yes, we agree, we have improved our discussion and brought in other reasons for the differences in seropositivity. 

It would have been interesting to have a discussion in relation to the findings of districts that have previously reported the cases, considering that some of these were sampled as part of this survey (e.g. Agago).

We did not see a difference between seropositivity and districts that reported human outbreaks 

Need to discuss the limitations/possible biases of the study/design and how this affects the conclusions. As well as further perspectives.

Purposive sampling could be one of the limitations of this study especially since sampling was biased against what we considered high-risk areas such as places of reported human outbreaks and ecological zones that favour tick vector survival. There is a need to design a follow-up study that is clearly random without bias towards regions where the disease is expected. Also, the assay used is an in-house assay that tends to underestimate the prevalence of CCHF as demonstrated by Balinandi et al, 2019. This has been added in the manuscript.

---

## [Decision Letter · Decision Letter 1]

10 Apr 2023

PONE-D-22-22276R1Seroepidemiological Investigation of Crimean Congo Hemorrhagic fever virus in Livestock in Uganda, 2017PLOS ONE

Dear Dr. Nyakarahuka,

Thank you for submitting your manuscript to PLOS ONE. After careful consideration, we feel that it has merit but does not fully meet PLOS ONE’s publication criteria as it currently stands. Therefore, we invite you to submit a revised version of the manuscript that addresses the points raised during the review process. The manuscript needs a revision according to the reviewer's comments.

We look forward to receiving your revised manuscript.

Kind regards,

Shawky M Aboelhadid, PhD

Academic Editor

PLOS ONE

Journal Requirements:

Reviewers' comments:

Reviewer's Responses to Questions

**Comments to the Author**

1. If the authors have adequately addressed your comments raised in a previous round of review and you feel that this manuscript is now acceptable for publication, you may indicate that here to bypass the “Comments to the Author” section, enter your conflict of interest statement in the “Confidential to Editor” section, and submit your "Accept" recommendation.

Reviewer #1: (No Response)

2. Is the manuscript technically sound, and do the data support the conclusions?

Reviewer #1: Yes

3. Has the statistical analysis been performed appropriately and rigorously? 

Reviewer #1: Yes

4. Have the authors made all data underlying the findings in their manuscript fully available?

Reviewer #1: No

5. Is the manuscript presented in an intelligible fashion and written in standard English?

Reviewer #1: Yes

6. Review Comments to the Author

Reviewer #1: Thanks for addressing the initial comments. The manuscript has been improved and reads well. Only minor comments:

Abstract:

1. In this sentence “CCHFV seropositivity appeared to be generally higher in northern districts of the country, though spatial trends among sampled districts were not obvious”, I suggest replacing ‘obvious’ to ‘examined’.

2. Make sure that decimals all along are the same, either 1 or 2. For example the total of the percentages of seropositive animals for all species does not add 100%. Please check this all along.

Methods:

1. “Herds were selected for sampling to be followed prospectively” – Not sure what this means, adding the word prospectively to the methods when this is a survey is confusing. Please clarify or remove.

2. “Assuming an average of 15 animals per herd, we 118 expected to sample 156 herds, distributed evenly throughout the 27 districts selected for sampling”- Ok, but it would be good to add how you selected the herds within each district and the 15 animals within each herd (when the herd had more than 15 animals). Add one sentence if each one of these steps was random or convenience sampling.

Results:

1. In table 1 please check that all the total of the animals for each variable is 3181. Some are not at the moment (e.g. Sex). Suggest to double check too in table 2 and 3, just in case.

2. “Holding all other variables in the model constant, the associations between CCHF seropositivity and animal breed and elevation were not statistically significant based on a 95% confidence limit, however, the estimated odds of seropositivity were lower among cross and exotic breeds compared to indigenous breeds, and the estimated odds of seropositivity among animals at lower elevations was higher than that among animals at higher elevations (Table 3)” – Don’t agree with what you say after 'however' because clearly you have your odds ratio indicating that based on your data there is no association. Suggesting otherwise might be misleading, maybe if you’d like to leave it better for a point of discussion.

Discussion

1. “However, Balinandi et al (2021) also performed serology on cattle samples using 278 a commercial CCHF serological testing kit, the ID screen CCHF double antigen multi-species 279 (IDVet), and found seropositivity of 75% in cattle (Balinandi, von Brömssen, et al., 2021; Sas et 280 al., 2018), thus suggesting some commercial assays may overestimate the true livestock 281 seroprevalence of CCHFV” – Still not sure if you can say this… not sure unless you’re sure it is the same area, same population and timing or if you have any evidence to support the claim that commercial tests overestimate CCHFv seroprevalence.

2. “We also found that the odds of seropositivity were higher among 315 animals sampled at lower elevations. Seroprevalence was also higher in animals with reported 316 stillbirth and abortion” – Agree but some of these aspects didn’t go to the multivariable model or didn’t come up as significant when all variables considered, for example elevation. I suggest that you make this distinction because sounds like you’d really like elevation to be one of the variables associated, but your risk model does not support that.

3. “However, as CCHF is not known to be symptomatic in animals, there is a 317 need to investigate further the impact of the CCHFV infection on animal production in terms of 318 reducing herd size and milk production. However, the interpretation and association of CCHFV 319 seropositivity with stillbirth and abortion could be confounding since the same animal populations 320 in Uganda are susceptible to other diseases that cause abortions such as brucellosis and Rift Valley 321 fever virus” – Double use of however, maybe rephrase.

4. “However, the interpretation and association of CCHFV 319 seropositivity with stillbirth and abortion could be confounding since the same animal populations 320 in Uganda are susceptible to other diseases that cause abortions such as brucellosis and Rift Valley 321 fever virus” – True, also the fact that this is a cross sectional study makes it impossible to test causation, only association.

5. “However, a stratified analysis limited to adult animals produced an odds ratio 336 approximately equal to that in the full analysis for female animals compared to male animals, 337 suggesting that age is not the primary explanation for the difference in seroprevalence between 338 male and female animals.” – Seems like adding results here, beware how you present this as it was not in your results.

7. PLOS authors have the option to publish the peer review history of their article (what does this mean?). If published, this will include your full peer review and any attached files.

Reviewer #1: No

While revising your submission, please upload your figure files to the Preflight Analysis and Conversion Engine (PACE) digital diagnostic tool, https://pacev2.apexcovantage.com/. PACE helps ensure that figures meet PLOS requirements. To use PACE, you must first register as a user. Registration is free. Then, login and navigate to the UPLOAD tab, where you will find detailed instructions on how to use the tool. If you encounter any issues or have any questions when using PACE, please email PLOS at figures@plos.org. Please note that Supporting Information files do not need this step.<quillbot-extension-portal></quillbot-extension-portal>

---

## [Author Response · Author response to Decision Letter 1]

25 May 2023

Reviewers' comments:

 Abstract:

1. In this sentence “CCHFV seropositivity appeared to be generally higher in northern districts of the country, though spatial trends among sampled districts were not obvious”, I suggest replacing ‘obvious’ to ‘examined’.

Thank you for the feedback. The recommended correction has been made

2. Make sure that decimals all along are the same, either 1 or 2. For example the total of the percentages of seropositive animals for all species does not add 100%. Please check this all along.

Methods:

1. “Herds were selected for sampling to be followed prospectively” – Not sure what this means, adding the word prospectively to the methods when this is a survey is confusing. Please clarify or remove.

2. “Assuming an average of 15 animals per herd, we 118 expected to sample 156 herds, distributed evenly throughout the 27 districts selected for sampling”- Ok, but it would be good to add how you selected the herds within each district and the 15 animals within each herd (when the herd had more than 15 animals). Add one sentence if each one of these steps was random or convenience sampling.

Clusters of herds were purposively selected based on specific criteria, including those with a high tick burden, high animal population and cooperative animal owners, those located in dry and wet areas, and those situated near international borders. Once the clusters were identified, herds with 15 or fewer animals were entirely sampled, while herds with more than 15 animals were sampled using proportional size sampling, where only 25% of the herd was selected for sampling. Individual animals were conveniently chosen and restrained in a crush, and blood samples were collected until the 25% achieved. This has been added in the manuscript. 

Results:

1. In table 1 please check that all the total of the animals for each variable is 3181. Some are not at the moment (e.g. Sex). Suggest to double check too in table 2 and 3, just in case.

Thanks for this. We have checked the tables for clarity 

2. “Holding all other variables in the model constant, the associations between CCHF seropositivity and animal breed and elevation were not statistically significant based on a 95% confidence limit, however, the estimated odds of seropositivity were lower among cross and exotic breeds compared to indigenous breeds, and the estimated odds of seropositivity among animals at lower elevations was higher than that among animals at higher elevations (Table 3)” – Don’t agree with what you say after 'however' because clearly you have your odds ratio indicating that based on your data there is no association. Suggesting otherwise might be misleading, maybe if you’d like to leave it better for a point of discussion.

Thanks for this observation. We have removed the last component 

Discussion

1. “However, Balinandi et al (2021) also performed serology on cattle samples using 278 a commercial CCHF serological testing kit, the ID screen CCHF double antigen multi-species 279 (IDVet), and found seropositivity of 75% in cattle (Balinandi, von Brömssen, et al., 2021; Sas et 280 al., 2018), thus suggesting some commercial assays may overestimate the true livestock 281 seroprevalence of CCHFV” – Still not sure if you can say this… not sure unless you’re sure it is the same area, same population and timing or if you have any evidence to support the claim that commercial tests overestimate CCHFv seroprevalence.

After careful consideration, we have concluded and acknowledged that we lack sufficient evidence to assert an overestimation of seropositivity by commercial ELISAs. As a result, we have made appropriate revisions to the statement, which now states:

Balinandi et al. (2021) performed serology tests on cattle samples in three of the studied districts using the ID screen CCHF double antigen multi-species (IDVet), a commercial CCHF serological testing kit. Their findings revealed a seropositivity rate of 75% in cattle. This suggests that the selection of ELISA assay could potentially result in variations in seropositivity rates(Balinandi, von Brömssen, et al., 2021; Sas et al., 2018).

2. “We also found that the odds of seropositivity were higher among 315 animals sampled at lower elevations. Seroprevalence was also higher in animals with reported 316 stillbirth and abortion” – Agree but some of these aspects didn’t go to the multivariable model or didn’t come up as significant when all variables considered, for example elevation. I suggest that you make this distinction because sounds like you’d really like elevation to be one of the variables associated, but your risk model does not support that.

We appreciate your astute observation, and as a result, we have revised the paragraph to provide an explanation for why we were unable to include these variables in the multivariable model due to missing data.

3. “However, as CCHF is not known to be symptomatic in animals, there is a 317 need to investigate further the impact of the CCHFV infection on animal production in terms of 318 reducing herd size and milk production. However, the interpretation and association of CCHFV 319 seropositivity with stillbirth and abortion could be confounding since the same animal populations 320 in Uganda are susceptible to other diseases that cause abortions such as brucellosis and Rift Valley 321 fever virus” – Double use of however, maybe rephrase.

This paragraph has been rephrased. 

4. “However, the interpretation and association of CCHFV 319 seropositivity with stillbirth and abortion could be confounding since the same animal populations 320 in Uganda are susceptible to other diseases that cause abortions such as brucellosis and Rift Valley 321 fever virus” – True, also the fact that this is a cross sectional study makes it impossible to test causation, only association.

This is correct observation. 

5. “However, a stratified analysis limited to adult animals produced an odds ratio 336 approximately equal to that in the full analysis for female animals compared to male animals, 337 suggesting that age is not the primary explanation for the difference in seroprevalence between 338 male and female animals.” – Seems like adding results here, beware how you present this as it was not in your results.

Thanks for this observation, this section has been edited out.

---

## [Editor Report · Decision Letter 2]

2 Jul 2023

Seroepidemiological Investigation of Crimean Congo Hemorrhagic fever virus in Livestock in Uganda, 2017

PONE-D-22-22276R2

Dear Dr. Luke,

We’re pleased to inform you that your manuscript has been judged scientifically suitable for publication and will be formally accepted for publication once it meets all outstanding technical requirements.

Kind regards,

Shawky M Aboelhadid, PhD

Academic Editor

PLOS ONE
---

## [Editor Report · Acceptance letter]

6 Jul 2023

PONE-D-22-22276R2 

Seroepidemiological Investigation of Crimean Congo Hemorrhagic fever virus in Livestock in Uganda, 2017 

Dear Dr. Nyakarahuka:

I'm pleased to inform you that your manuscript has been deemed suitable for publication in PLOS ONE. Congratulations! Your manuscript is now with our production department. 

Kind regards, 

on behalf of

Professor Shawky M Aboelhadid 

Academic Editor

PLOS ONE